

# Nuclear energy transition and $CO_2$ emissions nexus in 28 nuclear electricity-producing countries with different income levels

Haider Mahmood

Department of Finance, College of Business Administration, Prince Sattam bin Abdulaziz University, Alkharj, Saudi Arabia

## ABSTRACT

**Background:** Nuclear energy carries the least environmental effects compared to fossil fuels and most other renewable energy sources. Therefore, nuclear energy transition (NET) would reduce pollution emissions. The present study investigates the role of the NET on $CO_2$ emissions and tests the environmental Kuznets curve (EKC) in the 28 nuclear electricity-producing countries from 1996–2019.

**Methods:** Along with a focus on the whole panel, countries are divided into three income groups using the World Bank classification, *i.e.*, three Lower-Middle-Income (LMI), eight Upper-Middle-Income (UMI), and 17 High-Income (HI) countries. The cross-sectional dependence panel data estimation techniques are applied for the long and short run analyses.

**Results:** In the long run, the EKC is corroborated in HI countries' panel with estimated positive and negative coefficients of economic growth and its square variable. The Netherlands, Sweden, Switzerland, and the USA are found in the 2nd stage of the EKC. However, the remaining HI economies are facing 1st phase of the EKC. Moreover, economic growth has a monotonic positive effect on $CO_2$ emissions in LMI and UMI economies. NET reduces $CO_2$ emissions in UMI and HI economies. On the other hand, NET has an insignificant effect on $CO_2$ emissions in LMI economies. In the short run, the EKC is validated and NET has a negative effect on $CO_2$ emissions in HI countries and the whole panel. However, NET could not affect $CO_2$ emissions in LMI and UMI countries. Based on the long-run results, we recommend enhancing nuclear energy transition in UMI and HI economies to reduce $CO_2$ emissions. In addition, the rest of the world should also build capacity for the nuclear energy transition to save the world from global warming.

## INTRODUCTION

The electricity production from fossil fuels is majorly responsible for global greenhouse gas (GHG) emissions and global warming (*IPCC, 2013*). Energy production and consumption count for two-thirds of GHG emissions. $CO_2$ concentration increased by 408 parts per million (ppm) in 2018, which was 280 ppm before the industrial revolution. It resulted in

Corresponding author
Haider Mahmood,
haidermahmood@hotmail.com

an increasing 1.1 °C global average temperature during 2010–2019 compared to the pre-industrial level (*IAEA, 2020*). The Paris covenant on climate change is aimed to reduce global warming (*UNFCCC, 2015*). Switching to cleaner energy options, including nuclear and other renewable sources, would decrease global warming by reducing $CO_2$ emissions (*IEA, 2015*). Moreover, sustainable development goals (SDGs) also aim to improve global environmental quality by providing reasonable, reliable, and the latest energy resources, which may also reduce the environmental effects of economic activities for a better quality of life (*SDGs, 2021*). In this context, nuclear energy consumption (NEC) may promote sustainable development (*Uddin, 2019*), could save an economy from oil price fluctuations (*Lee & Chiu, 2011*), and would reduce the global warming issue as per the 2015 Paris Agreement (*IAEA, 2020*). Nuclear technology is a competitor of oil in electricity production after the oil crisis of the 1970s (*Toth & Rogner, 2006*) and NEC has the ability to replace fossil fuels effectively and quickly (*Sovacool, 2008*). In 2020, 10% of global electricity is produced from nuclear (*Pomponi & Hart, 2021*) and NEC covers around 4.3% of total primary energy demand (*BP, 2021a*).

NEC is almost free of $CO_2$ and other GHG emissions (*Rashad & Hammad, 2000*). For instance, one-kilowatt-hour electricity production from NEC releases emissions of 15 grams of $CO_2$ equivalent (*Saidi & Omri, 2020*), which is the least polluter compared to oil, gas, and coal consumption (*Weisser, 2007*). Moreover, NEC is also more efficient in reducing $CO_2$ emissions compared to other most renewable sources (*Wang et al., 2018*). For instance, nuclear power generation helped in reducing 74 Gt $CO_2$ during 1971–2018, which is equal to total emissions from all global power sectors during 2013–2018. During the last decade, nuclear power has avoided 2 Gt $CO_2$ annually, which is most efficient compared to other renewable sources except hydroelectricity (*IAEA, 2020*). Thus, NEC may reduce overall GHG emissions in any economy (*Vaillancourt et al., 2008*; *Sims, Rogner & Gregory, 2003*; *van der Zwaan, 2013*; *Goh & Ang, 2018*; *Jimenez & Flores, 2015*; *Adamantiades & Kessides, 2009*), which are majorly responsible for global warming (*Baek & Pride, 2014*). In another argument, literature claimed that NEC could help in decarbonizing the world due to its low-carbon technology. However, it can be responsible for nuclear accidents, radioactive waste, and pollution (*Fiore, 2006*; *Bandoc, 2018*). In response to this argument, *Sovacool & Monyei (2021)* estimated and found that replacing fossil fuel consumption with NEC has saved 42 lives from air-pollution-related deaths in China, India, the EU, and the US during 2000–2020. Hence, NEC could have a net positive effect on human lives.

Along with positive environmental effects, NEC also supports economic growth, which is called a growth hypothesis. This hypothesis states that increasing energy consumption would accelerate economic growth without a feedback effect. It attracts the attention of policymakers and researchers in the last two decades and many researchers have tested this hypothesis. For instance, NEC accelerates economic growth in France (*Mbarek, Khairallah & Feki, 2015*; *Marques, Fuinhas & Nunes, 2016*), Japan, the UK, and the US (*Chu & Chang, 2012*), Pakistan (*Luqman, Ahmad & Bakhsh, 2019*; *Rehman et al., 2021*), India (*Wolde-Rufael, 2010*; *Heo, Yoo & Kwak, 2011*), Korea (*Yoo & Jung, 2005*; *Yoo & Ku, 2009*), Belgium and Spain (*Omri, Mabrouk & Sassi-Tmar, 2015*), Colombia, Peru, and Venezuela (*Ozturk,*

2017), 10 highest emitting countries (*Azam et al., 2021*) and Japan, the Netherlands, and Switzerland (*Wolde-Rufael & Menyah, 2010*). In an opposite direction, economic growth may increase energy consumption without a feedback effect, which is called a conservative hypothesis. In the same way, NEC could also serve the growing need for energy due to increasing economic growth. Many studies have corroborated this conservative hypothesis in their empirical exercises. For example, economic growth promotes NEC in the US (*Chu & Chang, 2012*), the UK (*Kirikkaleli, Adedoyin & Bekun, 2021*), Japan (*Lee & Chiu, 2011*), Bulgaria, Canada, the Netherlands, and Sweden (*Omri, Mabrouk & Sassi-Tmar, 2015*), France and Pakistan (*Yoo & Ku, 2009*), and Canada and Sweden (*Wolde-Rufael & Menyah, 2010*). Moreover, a feedback hypothesis explains a two-way relationship between energy consumption and economic growth. This hypothesis, in the relationship between NEC and economic growth, has been validated in Canada, Germany, and the UK (*Lee & Chiu, 2011*), France, the UK, Spain, and the US (*Wolde-Rufael & Menyah, 2010*), Switzerland (*Yoo & Ku, 2009*), and the USA, Pakistan, France, Brazil, and Argentina (*Omri, Mabrouk & Sassi-Tmar, 2015*). Furthermore, the neutrality hypothesis explains no relation between NEC and economic growth. Some studies have reported the validity of the neutrality hypothesis in the US (*Payne & Taylor, 2010*), the UK, Japan, Hungary, Finland, Switzerland, and India (*Omri, Mabrouk & Sassi-Tmar, 2015*), a panel of 18 countries (*Mbarek, Saidi & Amamri, 2018*), 11 out of 14 Organization for Economic Cooperation and Development (OECD) countries (*Nazlioglu, Lebe & Kayhan, 2011*), and Taiwan (*Wolde-Rufael, 2012*).

Another stream of literature has tested the NEC and pollutant emissions nexus in the panel of nuclear electricity-producing countries. NEC helped in reducing emissions in 25 countries (*Alam, 2013*), 18 OECD countries (*Lau et al., 2019*), 15 OECD countries (*Saidi & Omri, 2020*), 20 OECD countries (*Richmond & Kaufman, 2006*), 11 OECD countries (*Iwata, Okada & Samreth, 2012*), G-7 (*Nathaniel et al., 2021*), 12 countries (*Baek, 2015*), in BRICS (*Hassan et al., 2020*), 10 highest emitting countries (*Azam et al., 2021*), 16 countries (*Kim, 2021*), 10 countries (*Baek & Pride, 2014*), 18 countries contributing 95% nuclear reactors globally (*Lee, Kim & Lee, 2017*), nine countries (*Vo et al., 2020*), Europe and the globe (*Wagner, 2021*), and 19 countries (*Apergis et al., 2010*). On the other hand, some studies could not validate the effect of NEC on pollution emissions in the panel analyses (*Saidi & Ben Mbarek, 2016*; *Al-mulali, 2014*; *Sovacool et al., 2020*; *Jin & Kim, 2018*; *Pao & Chen, 2019*). In a single country analysis, literature found the negative effect of NEC on pollution emissions in India (*Danish, Ozcan & Ulucak, 2021*; *Syed, Kamal & Tripathi, 2021*), China (*Dong et al., 2018*; *Wang et al., 2018*), Korea (*Kim, 2020*), Iran (*Kargari & Mastouri, 2011*), Spain (*Pilatowska, Geise & Włodarczyk, 2020*), the US (*Baek, 2016*; *Menyah & Wolde-Rufael, 2010*), Israel (*Aslan & Cam, 2013*), and France (*Iwata, Okada & Samreth, 2010*; *Marques, Fuinhas & Nunes, 2016*). However, some studies provided opposite results and NEC increased pollution emissions in the US (*Pan & Zhang, 2020*), Pakistan (*Mahmood et al., 2020*), and South Africa (*Sarkodie & Adams, 2018*). However, NEC could not affect $CO_2$ emissions in Japan (*Ishida, 2018*). Along with testing NEC and pollution emissions nexus, the literature has also tested the environmental Kuznets curve (EKC) hypothesis. This hypothesis may be validated with an inverted U-shaped or an N-shaped relationship between economic growth and pollution emissions. A few studies

have tested and validated the EKC in nuclear electricity-producing countries (*Lee, Kim & Lee, 2017*; *Vo et al., 2020*; *Nathaniel et al., 2021*; *Danish, Ozcan & Ulucak, 2021*; *Dong et al., 2018*; *Iwata, Okada & Samreth, 2010*; *Sarkodie & Adams, 2018*; *Kim, 2021*). However, *Baek (2015)* could not validate the EKC in a panel of 12 high-income major nuclear-generating countries.

The present study contributes to the present state of literature by applying cross-sectional dependence (CD) in the estimation procedure of 28 nuclear electricity-producing countries, and by testing the effect of nuclear energy transition (NET) on $CO_2$ emissions. Some studies have cared about this issue in regressions analyses of limited sample nuclear countries, *i.e.*, BRICS and G-7 (*Hassan et al., 2020*; *Nathaniel et al., 2021*). On the other hand, some studies care about the CD in the causality analysis (*Azam et al., 2021*; *Lau et al., 2019*). Still, a comprehensive analysis is missing in the literature caring the CD issue in the model of the EKC testing for a maximum sample of nuclear electricity-producing countries. Ignoring CD analysis in a presence of statistically significant CD would generate biased and misleading results in the model (*Eberhardt, 2012*). In addition, most literature has used the NEC or NEC *per capita* to test the environmental effects of nuclear energy. Nowadays, nations are transforming their energy generation from nonrenewable to renewable sources. Therefore, the present study analyzes the effect of the NET variable, instead of NEC, on $CO_2$ emissions. Moreover, the present study analyzes a full panel of 28 nuclear electricity-producing countries and compares the three sub-samples of 17 high-income (HI), eight upper-middle-income (UMI), and three lower-middle-income (LMI) nuclear electricity-producing countries.

## METHODS

While talking about the determinants of $CO_2$ emissions, nobody can deny the role of economic growth. In addition, *Grossman & Krueger (1991)* argued and found that economic growth has a nonlinear effect on pollution emissions. It means that growth may increase emissions at a lower level of income and would reduce emissions at a higher level of income, which is called the EKC hypothesis (*Panayotou, 1993*). For example, economic growth surges with higher economic activities, energy consumption, and pollution emissions, which is called a scale effect (*Grossman & Krueger, 1995*; *Mahmood, 2022*). At an earlier stage of development, economies are focusing on economic growth ignoring the type of energy and energy efficiency issues. Later, economic growth may demand a cleaner environment for a better standard of living, and encourages investments in clean technologies, which generate technique and composition effects (*Komen, Gerking & Folmer, 1997*). The composition effect may alter the pattern of production from dirty to cleaner processes. On the other hand, the technique effect may promote cleaner technologies and/or energy efficiency in the production processes (*Khan et al., 2022*). In all of this journey, energy consumption would play a significant role in shaping the EKC hypothesis (*Shahbaz & Sinha, 2019*; *Dogan & Turkekul, 2016*; *Rahman, Nepal & Alam, 2021*; *Murshed, Haseeb & Alam, 2022*). Particularly, renewable energy and energy efficiency would play their role in shaping the EKC in the second phase of the EKC (*Murshed, Khan & Rahman, 2022*; *Alam et al., 2022*). Among the others, NEC would be

more helpful in shifting the economy from the first to the second phase of the EKC to enjoy the fruits of growth without harming the environment (*Danish, Ozcan & Ulucak, 2021*; *Murshed et al., 2022*). Therefore, the world has realized the importance of cleaner types of energy sources to save the environment from pollution (*IEA, 2015*). It would transform the energy demand from fossil fuels to nuclear and other renewable sources (*Hamid et al., 2022*). Accordingly, the study uses the Nuclear Energy Transition (NET) variable instead of a simple NEC variable. *Baek & Pride (2014)* proposed a simple model regressing the economic growth and NEC on $CO_2$ emissions. However, *Mahmood et al. (2020)* extended the model of *Baek & Pride (2014)* by adding the square term of the economic growth variable to test the EKC hypothesis. Following *Mahmood et al. (2020)* and using NET instead of NEC, our model is as follows:

$$CO_{it} = f(Y_{it}, Y_{it}^2, NET_{it}) \tag{1}$$

To have pleasant environmental effects of NEC, the ratio of nuclear to nonrenewable energy should increase. Therefore, $NET_{it}$ is defined as the natural log of the ratio of NEC to nonrenewable energy sources, *i.e.*, coal, gas, and oil consumption. $Y_{it}$ is the natural log of Gross Domestic Product (GDP) *per capita* and $Y_{it}^2$ is the square of $Y_{it}$. $CO_{it}$ is the natural log of $CO_2$ emissions in tons *per capita*. *i* represents 28 nuclear electricity-producing countries and *t* is a period from 1996–2019. Moreover, the sample countries are divided into 3 LMI, 8 UMI, and 17 HI countries, as mentioned in the appendix. The income classification of countries is done following the *World Bank (2021)*. The model, mentioned in Eq. (1), is applied to the whole panel of 28 countries and the three subgroups of 3 LMI, 8 UMI, and 17 HI countries' panels. Data on $CO_2$ emissions in million tons and data on oil, gas, coal, and NEC in exajoule are taken from *BP (2021b)*. Data of oil, gas, coal, and NEC help to develop the NET variable. Data on population and GDP *per capita* (constant 2010 US$) are taken from the *World Bank (2021)*.

In the panel data estimation, slope heterogeneity and cross-sectional dependence (CD) may be present in the model and would generate biased results (*Pesaran & Smith, 1995*; *Eberhardt, 2012*). Globalization connects the economies politically, socially, and environmentally. Moreover, international environment agreements force the global economies to adopt renewable sources, improve energy efficiency, and reduce dependence on nonrenewable energy. To follow the environmental targets, countries may apply environmental regulations at a different pace as per the capacity of the economies. Hence, cross-sectional dependence and slope heterogeneity may exist between the economic growth and pollution emissions relationship (*Menegaki, 2021*). Therefore, the slope heterogeneity test of *Pesaran & Yamagata (2008)* is employed. The cross-sectional dependence is tested by using the LM test of *Breusch & Pagan (1980)* and *Pesaran, Ullah & Yamagata (2008)*, and the CD test of *Pesaran (2021)*. *Breusch & Pagan (1980)* offered the following LM statistic to test the CD.

$$LM = T \sum_{i=1}^{N-1} \sum_{j=i+1}^{N} \widehat{\partial}_{ij}^2 \tag{2}$$

$\widehat{\partial}_{ij}^2$ is square of the pairwise correlation of residuals. Moreover, *Pesaran (2021)* offered the extension of the LM test to provide unbiased results for finite T and large N, which is as follows:

$$LM = \sqrt{\frac{2T}{N(N-1)}} \sum_{i=1}^{N-1} \sum_{j=i+1}^{N} \widehat{\partial}_{ij}^2 \qquad (3)$$

In addition, *Pesaran, Ullah & Yamagata (2008)* suggested another unbiased version of the LM test:

$$LM = \sqrt{\frac{2}{N(N-1)}} \sum_{i=1}^{N-1} \sum_{j=i+1}^{N} \widehat{\partial}_{ij} \frac{[(T-k)\widehat{\partial}_{ij}^2 - \mu_{Tij}]}{\sqrt{v_{Tij}^2}} \qquad (4)$$

$k$ is the number of regressors. $\mu_{Tij}$ and $v_{Tij}^2$ are the mean and variance of $[(T-k)\widehat{\partial}_{ij}^2]$, respectively. After testing the CD issue, the slope heterogeneity is tested by using the methodology of *Pesaran, Ullah & Yamagata (2008)* in the following way:

$$\tilde{\Delta} = \sqrt{N}\left[\frac{N^{-1}\tilde{S} - k}{\sqrt{2k}}\right] \qquad (5)$$

$\tilde{S}$ compares the estimated slopes from pooled OLS and fixed effects. In addition, *Pesaran & Yamagata (2008)* provided the biased-adjusted version of $\tilde{\Delta}$ as follows:

$$\tilde{\Delta}_{adj} = \sqrt{N}\left[\frac{N^{-1}S - E(\tilde{z}_{iT})}{\sqrt{Var(\tilde{z}_{iT})}}\right] \qquad (6)$$

$E(\tilde{z}_{iT})$ and $Var(\tilde{z}_{iT})$ are mean and variance. In the presence of CD and heterogeneity, traditional unit root tests cannot be applied. Hence, we use the cross-sectional augmented-Dickey-Fuller (CADF) test of *Pesaran (2007)*, which is given in the following equation:

$$\Delta y_{it} = \Omega_0 + \Omega_{1i} y_{it-1} + \Omega_{2i} \overline{y_{t-1}} + \Omega_{3i} \overline{\Delta Y_t} + e_{1it} \qquad (7)$$

$i$ shows countries, $t$ represents years, $\overline{y_{t-1}} = N^{-1} \sum_{i=1}^{N} y_{it-1}$ and $\overline{\Delta y_t} = N^{-1} \sum_{i=1}^{N} \Delta y_{it}$. Moreover, *Pesaran (2007)* suggests cross-sectional Im-Pesaran-Shin (CIPS) in following way:

$$CIPS = N^{-1} \sum_{i=1}^{N} CADF_i \qquad (8)$$

After testing the unit root, the *Westerlund (2007)* cointegration can be tested in the model. This cointegration test cares about cross-sectional dependence and heterogeneity in the model. The test statistics are as follows:

$$G_t = N^{-1} \sum_{i=1}^{N} \frac{\theta_i}{SE(\widehat{\theta}_i)} \qquad (9)$$

$$G_a = N^{-1} \sum_{i=1}^{N} \frac{T\theta_i}{1\acute{\theta}(1)} \tag{10}$$

$$P_t = \frac{\widehat{\theta}_i}{SE(\widehat{\theta}_i)} \tag{11}$$

$$P_a = T\hat{\theta} \tag{12}$$

In the above equations, $\hat{\theta}$ is an estimated coefficient of the differenced-lagged dependent variable on the differenced dependent variable in the error correction model framework. In the presence of slope heterogeneity, the traditional long-run estimates from fixed or random effects could not provide robust results. To care for slope heterogeneity in estimations, *Pesaran & Smith (1995)* suggested the mean group (MG) estimators. However, MG estimators may also be biased in the presence of cross-sectional dependence (*Eberhardt, 2012*). At first, *Pesaran (2006)* proposed the methodology of common correlated effects MG (CCEMG), which cares about cross-sectional dependence in the estimations. Later, *Kapetanios, Pesaran & Yamagata (2011)* extended the CCEMG in the following way:

$$CO_{it} = a_i + b_i f_t + c_{1i} Y_{it} + c_{2i} Y_{it}^2 + c_{3i} NET_{it} + d_{0i}\overline{CO_{it}} + d_{1i}\overline{Y_{it}} + d_{2i}\overline{Y_{it}^2} + d_{3i}\overline{NET_{it}} + e_{1it} \tag{13}$$

$c_{ji}$ are country-specific ($i$) coefficients of explanatory variables. The CCEMG estimates of explanatory variables can be calculated by averaging, $\hat{\beta} = N^{-1} \sum_{i=1}^{N} \hat{c}_i$. $f_t$ is an unobserved common factor. In addition, a methodology of *Eberhardt & Bond (2009)* is utilized, which is presented as follows:

$$CO_{it} = l_i + m_i f_t + n_{1i}\Delta Y_{it} + n_{2i}\Delta Y_{itt}^2 + n_{3i}\Delta NET_{it} + \sum_{t=2}^{T} o_i D_t + e_{2it} \tag{14}$$

$\Delta$ is difference operator and $D_t$ is time dummy. In Eq. (14), the augmented MG (AMG) estimates of explanatory variables can be estimated by averaging, $\hat{\mu} = N^{-1} \sum_{i=1}^{N} \hat{n}_i$. Moreover, *Chudik & Pesaran (2015)* and *Chudik et al. (2017)* proposed CD-autoregressive distributive lag (CD-ARDL) model as follows:

$$\begin{aligned} CO_{it} = g_i &+ \sum_{j=1}^{k1} q_{1ij} CO_{it-j} + \sum_{j=0}^{k2} h_{1ij} Y_{it-j} + \sum_{j=0}^{k3} h_{2ij} Y_{it-j}^2 + \\ &\sum_{j=j}^{k4} h_{3ij} NET_{it-j} + \sum_{j=0}^{k5} h_{4j}\overline{CO_{it}} + \sum_{j=0}^{k6} h_{5j}\overline{Y_{it}} + \sum_{j=0}^{k7} h_{6j}\overline{Y_{it}^2} + \\ &\sum_{j=0}^{k8} h_{7j}\overline{NET_{it}} + e_{3it} \end{aligned} \tag{15}$$

CD-ARDL estimates of explanatory variables can be estimated by averaging, $\hat{\pi} = \sum_{j=0}^{k} \hat{h}_{ij} / 1 - \sum_{j=0}^{k} \hat{q}_{ij}$.

## RESULTS AND DISCUSSION

Before formal data analyses, the graphical income distribution of LMI, UMI, and HI countries is presented in Fig. 1. Among the LMI countries, India and Pakistan are neighboring countries. However, Ukraine is located far away from India and Pakistan.

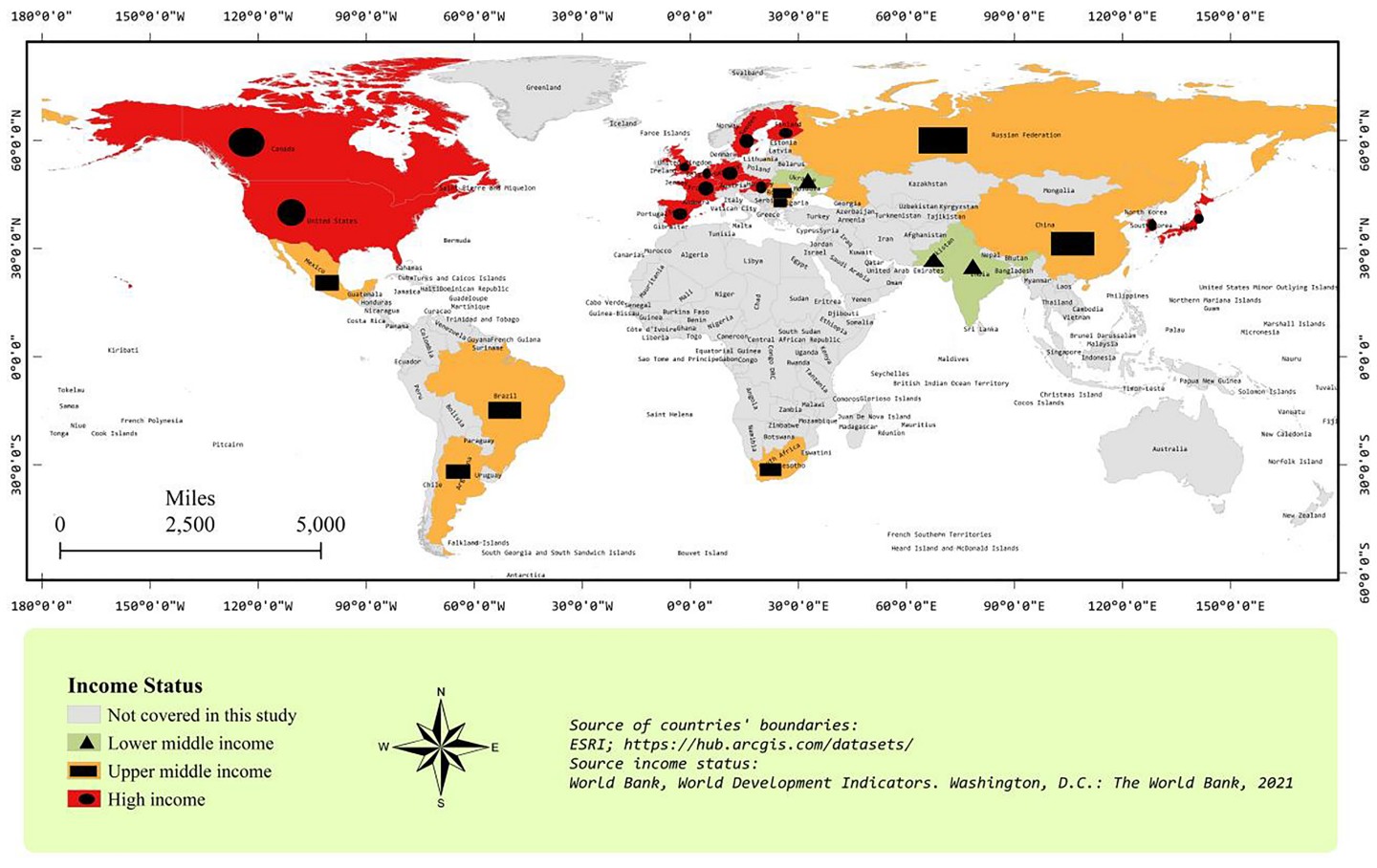

**Figure 1 Income group map.** Lower middle income country: Gross National Income *per capita* in current USD is between 1,036–4,045. Upper middle income country: Gross National Income *per capita* in current USD is between 4,046–12,535. High income country: Gross National Income *per capita* in current USD is more than 12,535 (ESRI, Redlands, CA, USA, https://hub.arcgis.com/datasets/; *World Bank, 2021*).

In the case of UMI countries, China and Russia are neighboring countries, Argentina and Brazil are neighboring countries, and Bulgaria and Romania are neighboring countries. However, these three neighboring pairs, Mexico, and South Africa are located far away from each other. In the case of HI countries, Canada and USA are neighboring countries and many European countries are neighbors as well. Figure 1 shows that all sample nuclear electricity-producing countries have a widespread distribution around the globe.

Figures 2 and 3 represent the geographical distribution of sample countries with respect to NET and $CO_2$ emissions *per capita*, respectively. The values of variables have been presented without a natural log to have a look at the original variables. Figure 2 shows that Canada and the USA are neighboring countries and are top-2 *per capita* $CO_2$ emitters. In the second-top group, we find Russia, South Korea, and some European countries. Most sample countries are in the third-top group of $CO_2$ emissions *per capita*, including China, Japan, South Africa, and some European countries. In Fig. 3, France and Sweden are in the highest NET group. Interestingly, most European countries are showing a higher level of NET compared to other sample countries. Moreover, the largest polluter countries, *i.e.*, the

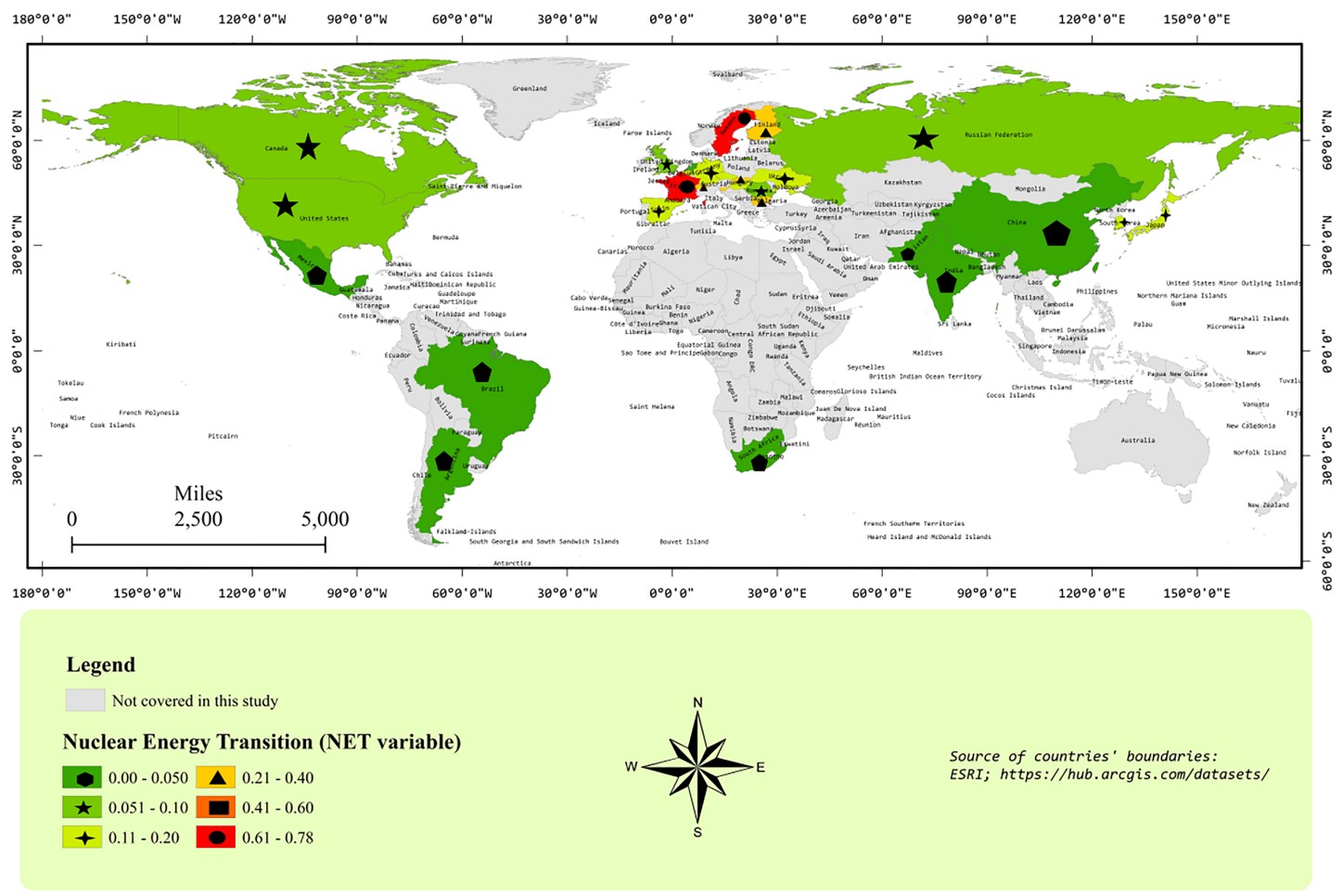

**Figure 2 CO₂ emission *per capita* map.** $CO_2$ emissions in tons *per capita* (ESRI, Redlands, CA, USA; https://hub.arcgis.com/datasets/).

USA, Canada, China, and Russia, have a low level of NET between 0–0.1. Figures 1–3 show interesting facts about the geographical distribution of nuclear electricity-producing countries. However, we ignore the spatial analyses in further estimations because of the widespread location of all sample countries around the Globe.

Table 1 shows descriptive statistics of variables in the full panel and the subpanels of three LMI, eight UMI, and 17 HI countries. The minimum value of $CO_2$ emissions is coming from LMI, and the maximum value of $CO_2$ emissions is coming from HI countries in the full panel. Thus, mean values show that higher-income countries are emitting higher emissions. The ratio of NET (0.912:1) is highest in HI countries in the HI panel and the full sample panel. However, NET is not the lowest in the LMI panel. It is because of a reason that the average NET ratio of Ukraine is 0.200:1 and the average NET ratio of Pakistan and India is approximately 0.012:1. Hence, the existence of Ukraine is showing a higher NET ratio in the LMI panel compared to the UMI panel. Otherwise, a higher level of income is mostly showing a higher NET on average in the targeted economies.
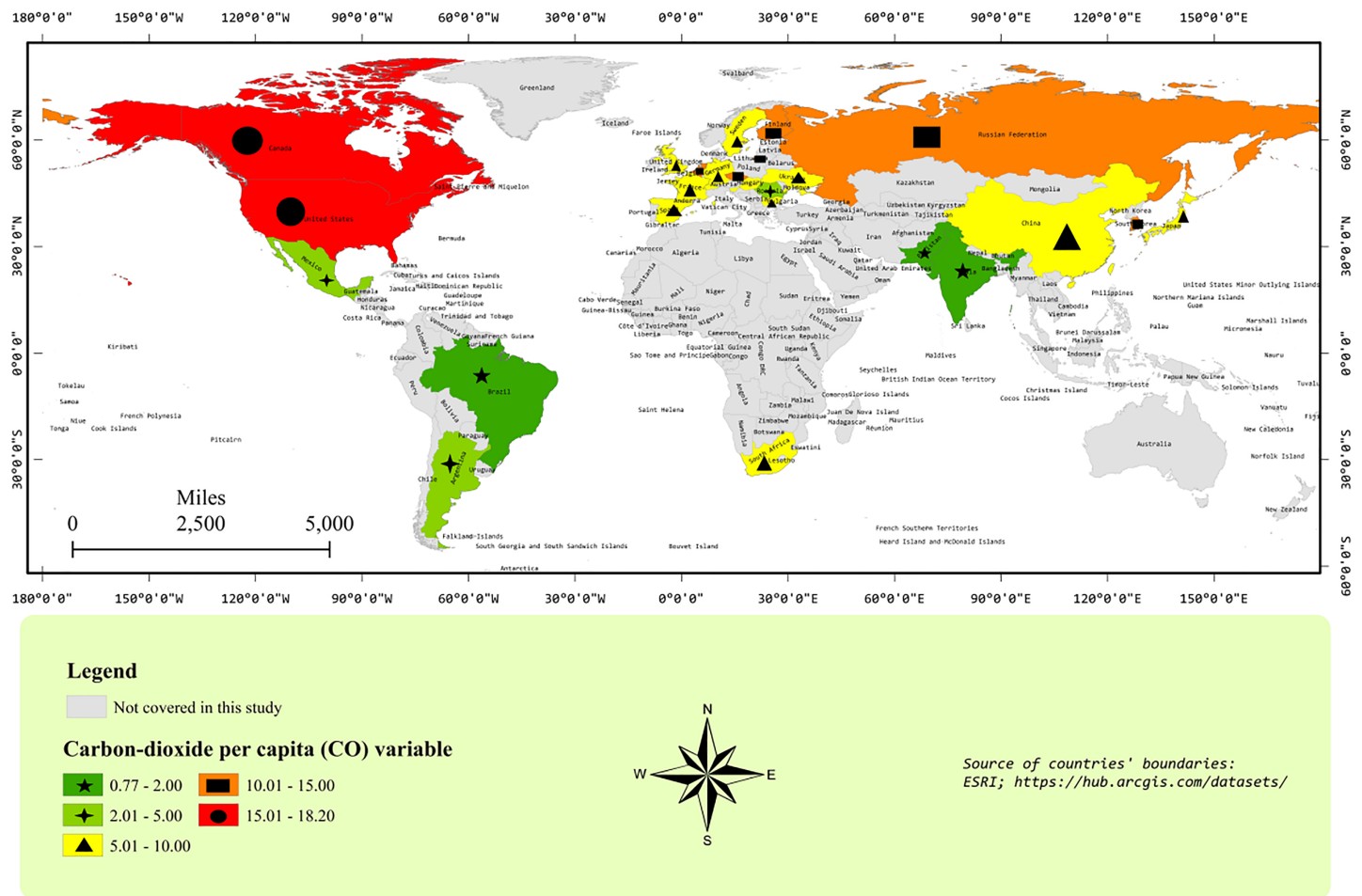

**Figure 3 The nuclear energy transition map.** Nuclear energy transition is defined as the ratio of nuclear energy consumption to nonrenewable energy sources, *i.e.*, coal, gas, and oil consumption (ESRI, Redlands, CA, USA; https://hub.arcgis.com/datasets/).

**Table 1 Descriptive statistics.**

| Income group | Variable | Observations | Mean | Standard deviation | Minimum | Maximum |
|---|---|---|---|---|---|---|
| Lower-Middle | $CO_{it}$ | 72 | 2.734 | 2.539 | 0.654 | 7.178 |
| | $Y_{it}$ | 72 | 1,627.714 | 851.212 | 711.929 | 3,322.005 |
| | $NET_{it}$ | 72 | 0.075 | 0.093 | 0.001 | 0.294 |
| Upper-Middle | $CO_{it}$ | 192 | 5.501 | 2.711 | 1.645 | 10.957 |
| | $Y_{it}$ | 192 | 7,883.353 | 2,558.503 | 1,332.350 | 12,122.610 |
| | $NET_{it}$ | 192 | 0.062 | 0.082 | 0.004 | 0.355 |
| High | $CO_{it}$ | 408 | 9.628 | 3.866 | 4.186 | 20.345 |
| | $Y_{it}$ | 408 | 37,582.440 | 15,901.790 | 8,992.874 | 83,093.190 |
| | $NET_{it}$ | 408 | 0.235 | 0.208 | <0.001 | 0.912 |
| Full | $CO_{it}$ | 672 | 7.104 | 4.258 | 0.654 | 20.345 |
| | $Y_{it}$ | 672 | 25,244.700 | 19,848.730 | 711.929 | 83,093.19 |
| | $NET_{it}$ | 672 | 0.167 | 0.190 | <0.001 | 0.912 |

**Note:**
$CO_{it}$, $Y_{it}$, and $NET_{it}$ represent $CO_2$ emissions in tons *per capita*, GDP *per capita*, and ratio of NEC to nonrenewable energy sources, respectively.

**Table 2 Cross dependence and slope heterogeneity tests.**

| Income group | Variable | Cross dependence | | | Slope heterogeneity | |
|---|---|---|---|---|---|---|
| | | Breusch-Pagan LM | Pesaran scaled LM | Pesaran CD | $\Delta$ | $\Delta_{adj}$ |
| Lower-Middle | $CO_{it}$ | 61.146 (0.000) | 15.919 (0.000) | 1.505 (0.186) | | |
| | $Y_{it}$ | 117.771 (0.000) | 32.265 (0.000) | 10.822 (0.000) | | |
| | $Y_{it}^2$ | 117.316 (0.000) | 32.134 (0.000) | 10.79 (0.000) | | |
| | $NET_{it}$ | 22.090 (0.001) | 4.645 (0.000) | 1.205 (0.228) | | |
| | Residual | 86.113 (0.000) | 23.127 (0.000) | 2.751 (0.006) | 7.976 (0.000) | 8.964 (0.000) |
| Upper-Middle | $CO_{it}$ | 209.316 (0.000) | 24.229 (0.000) | 5.687 (0.000) | | |
| | $Y_{it}$ | 532.763 (0.000) | 67.452 (0.000) | 22.986 (0.000) | | |
| | $Y_{it}^2$ | 532.287 (0.000) | 67.388 (0.000) | 22.979 (0.000) | | |
| | $NET_{it}$ | 161.311 (0.000) | 17.814 (0.000) | −0.439 (0.661) | | |
| | Residual | 153.5684 (0.000) | 16.7798 (0.000) | 3.3367 (0.000) | 10.448 (0.000) | 11.743 (0.000) |
| High | $CO_{it}$ | 1,988.773 (0.000) | 120.629 (0.000) | 33.819 (0.000) | | |
| | $Y_{it}$ | 2,570.097 (0.000) | 158.153 (0.000) | 50.527 (0.000) | | |
| | $Y_{it}^2$ | 2,567.667 (0.000) | 157.996 (0.000) | 50.602 (0.000) | | |
| | $NET_{it}$ | 485.524 (0.000) | 23.595 (0.000) | 3.023 (0.0025) | | |
| | Residual | 1,804.863 (0.000) | 108.757 (0.000) | 37.341 (0.000) | 18.502 (0.000) | 20.794 (0.000) |
| Full | $CO_{it}$ | 4,436.846 (0.000) | 147.619 (0.000) | 15.626 (0.000) | | |
| | $Y_{it}$ | 7,603.202 (0.000) | 262.778 (0.000) | 86.868 (0.000) | | |
| | $Y_{it}^2$ | 7,591.504 (0.000) | 262.353 (0.000) | 86.801 (0.000) | | |
| | $NET_{it}$ | 1,794.946 (0.000) | 51.834 (0.000) | 3.075 (0.0021) | | |
| | Residual | 5,064.066 (0.000) | 170.431 (0.000) | 41.256 (0.000) | 25.025 (0.000) | 28.126 (0.000) |

**Note:**

$CO_{it}$, $Y_{it}$, and $NET_{it}$ represent $CO_2$ emissions in tons *per capita*, GDP *per capita*, and ratio of NEC to nonrenewable energy sources, respectively. All variables are utilized in natural logarithm form.

Before proceeding with regression analysis, CD and slope-heterogeneity were tested to ensure unbiased conclusions from regression analyses. Results of CD tests in Table 2 show that the null hypothesis is rejected in the full panel and all subpanels of nuclear countries in the case of all variables, except $CO_t$ and $NET_t$ in lower-middle countries as per Pesaran CD results. Moreover, CD tests also corroborate the cross-sectional dependence in residuals of regressions in the full panel and all subpanels of nuclear countries. Hence, we get sufficient evidence to include CD in further analyses. In addition, the slope heterogeneity test rejects the null hypothesis in the full panel and all sub-groups of nuclear countries. Therefore, we care about slope heterogeneity and CD in further analyses.

CD tests suggest CD unit root analyses. Hence, we apply CADF and CIPS tests and the results are presented in Table 3. Results show that all panel series are non-stationary at the level in all countries' subgroups and the full panel. On the other hand, all variables are stationary at their first differences in all panel subgroups and the full panel. Hence, the order of integration is one in all panel subgroups and the full panel. So, we may move for

**Table 3 Panel unit root analyses.**

| Variable | Lower-Middle | | Upper-Middle | | High | | Full | |
|---|---|---|---|---|---|---|---|---|
| | C | C & T | C | C & T | C | C & T | C | C & T |
| CADF test at level | | | | | | | | |
| $CO_{it}$ | −1.632 | −2.641 | −1.496 | −1.657 | −1.780 | −2.412 | −1.733 | −2.504 |
| $Y_{it}$ | −1.168 | −2.264 | −1.454 | −2.006 | −1.867 | −2.371 | −1.910 | −2.281 |
| $Y_{it}^2$ | −1.045 | −2.242 | −1.445 | −2.229 | −1.814 | −2.329 | −1.896 | −2.227 |
| $NET_{it}$ | −1.558 | −1.386 | −1.560 | −2.192 | −1.423 | −2.036 | −1.687 | −2.017 |
| CIPS Test at level | | | | | | | | |
| $CO_{it}$ | −2.183 | −2.233 | −2.155 | −2.092 | −1.650 | −1.488 | −0.904 | −2.556 |
| $Y_{it}$ | −1.395 | −2.027 | −1.989 | −2.340 | −1.880 | −1.562 | −1.615 | −1.862 |
| $Y_{it}^2$ | −1.301 | −2.043 | −1.961 | −2.455 | −1.934 | −1.805 | −1.655 | −1.772 |
| $NET_{it}$ | −1.515 | −2.174 | −1.753 | −1.847 | −1.440 | −2.581 | −1.898 | −1.629 |
| CADF test at first difference | | | | | | | | |
| $\Delta CO_{it}$ | −3.463*** | −3.787*** | −2.423** | −2.913** | −2.660*** | −2.996*** | −2.289*** | −3.624*** |
| $\Delta Y_{it}$ | −3.140*** | −3.829*** | −3.104*** | −3.143*** | −2.486*** | −2.757** | −2.952*** | −2.669** |
| $\Delta Y_{it}^2$ | −3.335*** | −3.733*** | −3.114*** | −3.098*** | −2.448*** | −2.744*** | −2.423*** | −2.596*** |
| $\Delta NET_{it}$ | −3.083*** | −3.305*** | −2.625*** | −3.217*** | −2.513*** | −2.618*** | −3.401*** | −3.430*** |
| CIPS Test at first difference | | | | | | | | |
| $\Delta CO_{it}$ | −3.511*** | −3.462*** | −4.164*** | −4.342*** | −4.961*** | −5.013*** | −3.918*** | −4.375*** |
| $\Delta Y_{it}$ | −3.858*** | −3.957*** | −2.989*** | −3.340*** | −3.057*** | −3.525*** | −3.159*** | −3.517*** |
| $\Delta Y_{it}^2$ | −3.749*** | −3.945*** | −2.763*** | −2.970** | −3.009*** | −3.479*** | −3.109*** | −3.479*** |
| $\Delta NET_{it}$ | −3.582*** | −3.477*** | −5.186*** | −5.364*** | −5.201*** | −5.393*** | −4.029*** | −5.277*** |

Notes:

Two asterisks (**) and three asterisks (***) show stationary at 5% and 1% level of significance, respectively.

$\Delta$ is a first difference operator. $CO_{it}$, $Y_{it}$, and $NET_{it}$ represent $CO_2$ emissions in tons *per capita*, GDP *per capita*, and ratio of NEC to nonrenewable energy sources, respectively. All variables are utilized in natural logarithm form.

**Table 4 Westerlund cointegration test.**

| Test stat | Lower-Middle | Upper-Middle | High | Full |
|---|---|---|---|---|
| Gt | −7.745 (0.000) | −2.175 (0.957) | −6.974 (0.000) | −5.5214 (0.000) |
| Ga | −1.210 (1.000) | −15.521 (0.000) | −18.524 (0.000) | −16.291 (0.000) |
| Pt | −2.203 (0.988) | −3.209 (1.000) | −12.352 (0.000) | −8.524 (0.085) |
| Pa | −1.328 (0.998) | −2.103 (1.000) | −25.631 (0.000) | −21.922 (0.000) |

Note:

Gt, Ga, Pt, and Pa are test statistics of Westerlund's panel cointegration test.

cointegration analyses. In Table 4, the cointegration is verified in the high-income panel and the full panel by rejecting the null hypothesis of no-cointegration in all four statistics of Westerlund test. Moreover, cointegration is found in the lower-middle-income panel with significant Gt statistics and the upper-middle-income panel with significant Ga statistics. Thus, we may claim for cointegration in all models and can proceed with regression analyses.

Table 5 shows the CD-ARDL, CCEMG, and AMG results in all four panels. CCEMG and AMG results are reported to verify the robustness of CD-ARDL results. We may

**Table 5 Regression analyses.**

| Technique | Variable | Lower-Middle | Upper-Middle | High | Full |
|---|---|---|---|---|---|
| CD-ARDL | Long run | | | | |
| | $Y_{it}$ | 23.634 (0.027) | 9.280 (0.052) | 14.576 (0.028) | 10.902 (0.095) |
| | $Y_{it}^2$ | −1.146 (0.209) | −0.445 (0.458) | −0.671 (0.031) | −0.506 (0.105) |
| | $NET_{it}$ | −0.073 (0.233) | −0.001 (0.095) | −0.104 (0.000) | −0.085 (0.000) |
| | Short run | | | | |
| | $Y_{it}$ | 43.174 (0.643) | 15.257 (0.770) | 28.732 (0.023) | 21.968 (0.074) |
| | $Y_{it}^2$ | −2.083 (0.217) | −0.726 (0.491) | −1.324 (0.025) | −1.023 (0.080) |
| | $NET_{it}$ | −0.139 (0.260) | −0.013 (0.684) | −0.206 (0.000) | −0.172 (0.000) |
| | $ECT_{it-1}$ | −0.660 (0.000) | −0.781 (0.000) | −0.940 (0.000) | −0.898 (0.000) |
| CCEMG | $Y_{it}$ | 4.031 (0.086) | 6.910 (0.044) | 18.6556 (0.046) | 10.493 (0.038) |
| | $Y_{it}^2$ | −0.238 (0.195) | −0.302 (0.583) | −0.860 (0.051) | −0.547 (0.032) |
| | $NET_{it}$ | −0.027 (0.243) | −0.032 (0.037) | −0.132 (0.000) | −0.099 (0.000) |
| AMG | $Y_{it}$ | 1.820 (0.099) | 20.256 (0.056) | 19.995 (0.057) | 7.090 (0.086) |
| | $Y_{it}^2$ | −0.065 (0.378) | −1.040 (0.267) | −0.927 (0.068) | −0.320 (0.123) |
| | $NET_{it}$ | −0.019 (0.279) | −0.052 (0.000) | −0.150 (0.000) | −0.112 (0.000) |

**Note:**
$CO_{it}$, $Y_{it}$, and $NET_{it}$ represent $CO_2$ emissions in tons *per capita*, GDP *per capita*, and ratio of NEC to nonrenewable energy sources, respectively. All variables are utilized in natural logarithm form.

conclude the findings from the CD-ARDL technique because of its superiority over other techniques. However, we report all results for completeness. Long-run results show that the coefficients of $Y_t$ and $Y_t^2$ are positive and statistically insignificant in LMI and UMI countries' panels and the whole panel. Hence, economic growth has a monotonic positive impact on emissions in LMI, UMI and the whole panel. Therefore, the EKC hypothesis is not corroborated in these subpanels and the whole panel. On the other hand, the EKC is validated with the positive and negative effects of $Y_t$ and $Y_t^2$ in the HI panel with a turning point of 52,033 US dollars. The turning point is calculated from coefficients of $Y_t$ and $Y_t^2$ in the HI countries' panel CD-ARDL results, using the formula [exponent of −14.5758/2 (−0.6711)]. As per the high-income countries' turning point, the Netherlands, Sweden, Switzerland, and the USA are found in the second phase of the EKC. However, the rest countries are in the first phase of the EKC. It shows that NET helps these HI countries to shift their economies in the second phase of the EKC to enjoy the positive environmental consequences of economic growth. In the NEC-related studies, the EKC has been corroborated in the panel of G-7 high-income countries (*Nathaniel et al., 2021*) and a panel of 16 countries with a mixed level of income (*Kim, 2021*). However, *Baek (2015)* could not validate the EKC in a panel of 12 high-income countries. Moreover, some studies confirm the EKC in country-specific analysis (*Lee, Kim & Lee, 2017*; *Vo et al., 2020*; *Danish, Ozcan & Ulucak, 2021*; *Dong et al., 2018*; *Iwata, Okada & Samreth, 2010*; *Sarkodie & Adams, 2018*).

The nuclear energy transition ($NET_{it}$) reduces emissions in UMI, HI, and the full panel. The empirical literature has also corroborated that NEC reduced emissions in the panel of nuclear-producing countries (*Iwata, Okada & Samreth, 2012*; *Alam, 2013*; *Lau et al., 2019*;

*Saidi & Omri, 2020*; *Hassan et al., 2020*; *Azam et al., 2021*; *Kim, 2021*; *Baek & Pride, 2014*; *Vo et al., 2020*; *Wagner, 2021*; *Apergis et al., 2010*). In a comparison, $NET_{it}$ has a greater magnitude of effect in HI countries compared to UMI countries. On average, HI countries have a higher level of NET compared to UMI countries, as shown in Table 1 and Fig. 3. It helped the HI countries to reduce $CO_2$ emissions to a greater extent compared to UMI countries. On the other hand, $NET_{it}$ has statistically insignificant effects on emissions in LMI countries. Hence, our results show that NET could not affect $CO_2$ emission in LMI economies. This result is natural because the ratio of nuclear to fossil fuel consumption is lesser than 0.02 in most lower-middle-income countries. The insignificant effect of NEC on pollution emissions is reported in some empirical studies (*Al-mulali, 2014*; *Sovacool et al., 2020*; *Jin & Kim, 2018*; *Pao & Chen, 2019*; *Saidi & Ben Mbarek, 2016*).

The short-run results are displayed in Table 5. The coefficients of $ECT_{t-1}$ are negative and statistically significant in all estimated panels. Economic growth and $NET_{it}$ have statistically insignificant effects on $CO_2$ emissions in LMI and UMI countries. However, the EKC is validated in HI countries and the full panel with turning points of 51,632 USD dollars [exponent of $-28.7315/2(-1.3238)$] and 46,076 USD dollars [exponent of $-21.9679/2(-1.0229)$], respectively. As per the short-run result of the turning point of HI countries, the Netherlands, Sweden, Switzerland, and the USA are found in the second phase of the EKC. The $NET_{it}$ negatively affects emissions in the HI panel and the full panel. Moreover, the short-run coefficients are greater than the long-run estimates. Hence, the nuclear energy transition helps in reducing $CO_2$ emissions in a greater amount in the short run compared to the long run.

## CONCLUSIONS

The nuclear energy transition could help in reducing pollution emissions. Hence, we tested the effect of the NET on $CO_2$ emissions in the 28 nuclear electricity-producing countries from 1996–2019. We utilized the full panel of 28 countries, and the subpanels of 17 HI countries, eight UMI countries, and three LMI countries. Further, we also test the EKC hypothesis. For this purpose, we utilize the CD panel techniques because CD is presented in all investigated panels. Cross-sectional dependence was validated through various CD tests. Moreover, the order of integration is one in unit root analyses and cointegration was corroborated in all investigated panels. The long and short results are estimated through CD-ARDL. The robustness of the long-run results is tested by CCEMG and AMG estimates. The major conclusions remain the same with all estimation techniques. In the long run, economic growth shows a monotonic positive impact on emissions in LMI and UMI countries' panels and the whole panel. Hence, economic growth degrades the environment. However, economic growth and its square term have positive and negative effects on $CO_2$ emissions in HI countries. Therefore, the EKC is corroborated in high-income nuclear electricity-producing countries with a turning point of 52,033 US dollars in the long run and 51,632 US dollars in the short run. As per constant GDP *per capita*, the Netherlands, Sweden, Switzerland, and the USA are found in the second phase of the EKC in both the long and short run. Thus, economic growth helps in reducing $CO_2$ emissions in these economies. However, the rest of the analyzed HI economies are at the

first phase of the EKC. So, the economic growth of these economies could have environmental consequences because of increasing $CO_2$ emissions. NET has a statistically insignificant effect on $CO_2$ emissions in the LMI panel and has a negative effect on emissions in UMI, HI, and the whole panel. Moreover, the magnitude of the effect of NET is higher in the HI panel compared to the UMI panel. NET was captured through the ratio of nuclear to fossil fuels energy consumption. On average, the HI countries have a higher level of NET compared to UMI countries. Hence, the increasing dependence on nuclear energy in the total energy mix of HI countries has helped in reducing $CO_2$ emissions to a greater extent compared to UMI countries. In the short-run results, NET has also a negative effect on $CO_2$ emissions in HI countries and the full panel. However, NET could not affect $CO_2$ emissions in LMI and UMI countries. We recommend increasing the nuclear power share in the total energy mix of UMI and HI nuclear electricity-producing countries. Moreover, LMI nuclear electricity-producing countries should also enhance the nuclear electricity production capacity to have a positive environmental effect of the NET. In addition, the rest of the world, other than nuclear power producers, should also install nuclear plants for electricity production to improve their environmental condition, which would help in reducing pollution emissions and global warming as per the Paris Agreement.

## ACKNOWLEDGEMENTS

I thank the anonymous reviewers and academic editor for their valuable comments.

### Funding

The author received no funding for this work.

### Competing Interests

Haider Mahmood is an Academic Editor for PeerJ.

### Author Contributions

- Haider Mahmood conceived and designed the experiments, performed the experiments, analyzed the data, prepared figures and/or tables, authored or reviewed drafts of the article, and approved the final draft.

### Data Availability

The raw data are available in the Supplemental Files.

### Supplemental Information

Supplemental information for this article can be found online at http://dx.doi.org/10.7717/peerj.13780#supplemental-information.

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
