# Peer review of "Nuclear energy transition and CO2 emissions nexus in 28 nuclear electricity-producing countries with different income levels"

_PeerJ, doi:10.7717/peerj.13780_

## Round 0.1 · original submission · Major Revisions

Dear Dr. Mahmood,
Thank you very much for submitting your manuscript to PeerJ. As you shall see, the reviewers have now commented on your manuscript and are suggesting a Major Review. Please go through their comments and revise your manuscript according. Please submit point by point response rebuttal to their comments.

Further, there are some suggestions from my side also.

1. Overall, the work is excellent. However, the element of visual understanding of the results from this manuscript is lacking. I strongly suggest presenting all your results in Map Representations (GIS-based), which would add massive value to your work.

2. Please add footnotes to each table describing each variable abbreviation. I hope you have understood my concern.

I am looking forward to receiving your revised manuscript as soon as possible.

Thanks and best regards

Gowhar Meraj

Reviewer 1 ·

Basic reporting

I have read the manuscript thoroughly and came to a conclusion that the study can be considered for publication after incorporating the following issues:
The abstract section can have space to improve adding short-run results and improving policy discussions.
The introduction needs to be improved by clearly presenting the contributions of the study after discussing the literature gap.
The reasons to use the cross-sectional dependence techniques should be mentioned in the method section.

Experimental design

No comments

Validity of the findings

No comments

Additional comments

The discussions of the findings can be extended.
Limitations of the study and future direction should be provided in the conclusion section.

Reviewer 2 ·

Basic reporting

Good

Experimental design

Excellent

Validity of the findings

Very Good

Additional comments

The title “Nuclear energy transition and CO2 emissions nexus in 28 nuclear electricity-producing countries with different income levels: Evidence from cross-sectional dependence panel techniques” is an interesting topic and is a good contribution to energy and environment literature as nuclear energy transition would help reduce global warming. However, the manuscript needs a revision before the next round of the review process. The following points would helpfully help improve the current version of the paper:
The background in the abstract section needs more explanations to convey how nuclear energy would reduce pollution emissions. The method in the abstract section should provide the number of countries in subgroups of the whole sample and the criteria for such differentiation. The results in the abstract section should explain which results of the study validate the existence of the EKC. Moreover, I could see only long-run results in the abstract and the short-run results should also be provided in the abstract section, so the abstract section independently conveys the summary of the whole article.
The statement “Energy production and consumption count for two-thirds of GHG emissions” in the introduction section need supporting reference.
In the third paragraph, three hypotheses are named in the relationship between economic growth and nuclear energy consumption. However, the hypothesis of no-relationship is not explicitly named in lines 89-93, which needs to be provided.
Second last paragraph of the introduction section smartly explains the positive, negative, and statistically insignificant effects of nuclear energy on pollution emissions in literature. However, the results of the existence and non-existence of the EKC in lines 112-116 are missing. It will be very interesting to know how many studies could validate the EKC in nuclear energy-producing countries. Hence, the literature quoted on the testing of the EKC should be classified into existence and non-existence of the EKC in the nuclear energy-producing countries.
The theoretical background of the EKC hypothesis is well explained in the first paragraph of the method section with classical studies. However, some latest references on scale, composition, and technique effects should also be cited. Moreover, there is a need to add more latest references to comprehend the role of nuclear and renewable energy and energy efficiency in shaping the EKC in the second phase. For this purpose, the following articles may be helpful:
https://doi.org/10.1016/j.egyr.2022.03.138
https://doi.org/10.1016/j.renene.2022.05.065
https://doi.org/10.1016/j.renene.2022.03.121
In the second paragraph of the method section, the number of nuclear electricity-producing countries is not mentioned. Moreover, the model has also been replicated in the case of the subgroup of the whole sample, which needs to be discussed in this paragraph as well.
How is the turning point achieved from the calculation in the results section? The formula for this calculation should be mentioned in interpretations for more clarity. Moreover, the turning points of the EKC are provided for both the long and short run. However, the countries in the second stage are only mentioned as per long-run results and are missing in the case of short-run results. Therefore, the countries in the second phase of the EKC should also be mentioned in both sections of interpretation and conclusion. Moreover, it would help the reader and reviewer if tables are provided in the text instead of provided at the end of the manuscript.

·

Basic reporting

I have reviewed the paper titled “Nuclear energy transition and CO2 emissions nexus in 28 nuclear electricity-producing countries with different income levels: Evidence from cross-sectional dependence panel techniques”. It is a fine attempt and investigates a very interesting topic, which has been explored relatively lesser in the empirical literature in the context of nuclear producing countries. Therefore, it is a reasonable contribution to the energy and nuclear producing countries ‘environmental literature. Moreover, the methods of cross sectional dependence panel unit root test, cointegration test, and regression analyses, utilized in this study, are also advanced and appropriate as per the sample of the study and as per trends of recent research in this area. However, the following comments may add to the quality of the paper:
1. Title of the study is very long, which may limit to “Nuclear energy transition and CO2 emissions nexus in 28 nuclear electricity-producing countries with different income levels”.
2. The introduction section is well written in the context of nuclear energy, economic growth and pollutant emissions relationships. However, we cannot ignore the discussions of other renewable energy sources in determining pollution emissions. Therefore, the discussions on renewable energy use and pollution emissions nexus may be added and the following recent references may support the discussions on the renewable energy consumption and pollution emissions nexus:
-: Murshed, M., Rahman, M. A., Alam, M. S., Ahmad, P., & Dagar, V. (2021). The nexus between environmental regulations, economic growth, and environmental sustainability: linking environmental patents to ecological footprint reduction in South Asia. Environmental Science and Pollution Research, 28, 49967 - 49988.
-: Rehman, A., Ma, H., Radulescu, M., Sinisi, C. I., Păunescu, L. M., Alam, M. S., & Alvarado, R. (2021). The Energy Mix Dilemma and Environmental Sustainability: Interaction among Greenhouse Gas Emissions, Nuclear Energy, Urban Agglomeration, and Economic Growth. Energies.
-: Murshed, M., Haseeb, M., & Alam, M. S. (2021). The Environmental Kuznets Curve hypothesis for carbon and ecological footprints in South Asia: the role of renewable energy. GeoJournal, 1-28.
-: Murshed, M., & Alam, M. S. (2021). Estimating the macroeconomic determinants of total, renewable, and non-renewable energy demands in Bangladesh: the role of technological innovations. Environmental Science and Pollution Research, 28, 30176 - 30196.
-: Hamid, I., Alam, M. S., Kanwal, A., Jena, P. K., Murshed, M., & Alam, R. (2022). Decarbonization pathways: the roles of foreign direct investments, governance, democracy, economic growth, and renewable energy transition. Environmental Science and Pollution Research, 1 - 16.
-: Alam, M. S., Alam, M. N., Murshed, M., Mahmood, H., & Alam, R. (2022). Pathways to securing environmentally sustainable economic growth through efficient use of energy: a bootstrapped ARDL analysis. Environmental Science and Pollution Research, 1-15.
-: Alam, Q., Alam, S., & Jamil, S. A. (2016). Oil demand and price elasticity of energy consumption in the GCC countries: A panel cointegration analysis. Business and Economic Horizons (BEH), 12(1232-2017-2390), 63-74.

3. The discussions on the relationship between nuclear energy consumption and economic growth relationship may be extended by debating the four hypotheses mentioned in the third paragraph of the introduction section.
4. In line with the above comment, in the introduction section, the discussion on the Environmental Kuznets Curve may be extended with rules of verification of the validity of the Environmental Kuznets Curve hypothesis and with the empirical support in favor of the hypothesis and evidence in the opposing views in the context of the nuclear energy producing countries as per established literature in the context of nuclear energy producing countries.
5. The discussion on the Environmental Kuznets Curve may also be enhanced in the method section, particularly in the context of the scale effect, technique effect, and composition effect of the energy transition on pollution emissions.
6. Interpretation of the results should also be increased, particularly, in the context of the validity of the Environmental Kuznets Curve hypothesis and implications of the long run and the short run results in the context of the investigated countries.

Experimental design

The methods of cross sectional dependence panel unit root test, cointegration test, and regression analyses, utilized in this study, are also advanced and appropriate as per the sample of the study and as per trends of recent research in this area.

Validity of the findings

Ok

Additional comments

Accepted with minor revision

---

## Round 0.2 · accepted · Accept

Thank you for considering all the suggestions made by the reviewers and myself. The map visualizations are very good now. They depict what I intended from your study. Thank you again. Best of luck.

Reviewer 2 ·

Basic reporting

It has been improved.

Experimental design

I was fine before and even has been improved by incorporating my comments.

Validity of the findings

The validity of the findings is quite satisfactory with the applied and reported alternative robustness techniques.

Additional comments

All of my raised concerns in the first round of review have been resolved in this revised version. The paper has been improved and may be accepted in this current form.

·

Basic reporting

Author addressed all valid points.

Experimental design

Ok

Validity of the findings

Ok

Additional comments

Accepted without any changes